Subject Area:
biophysics/cellular biology/microbiology/
structural biology/immunology

Keywords:
adenovirus, infection, cell networks,
virus entry, capsid, uncoating

Author for correspondence:
Justin W. Flatt
e-mail: justin.flatt@helsinki.fi

# Adenovirus flow in host cell networks

## Justin W. Flatt and Sarah J. Butcher

Faculty of Biological and Environmental Sciences and HiLIFE-Institute of Biotechnology, University of Helsinki, 00790 Helsinki, Finland

  JWF, 0000-0001-5406-8156; SJB, 0000-0001-7060-5871

Viruses are obligatory parasites that take advantage of intracellular niches to replicate. During infection, their genomes are carried in capsids across the membranes of host cells to sites of virion production by exploiting cellular behaviour and resources to guide and achieve all aspects of delivery and the downstream virus manufacturing process. Successful entry hinges on execution of a precisely tuned viral uncoating program where incoming capsids disassemble in consecutive steps to ensure that genomes are released at the right time, and in the right place for replication to occur. Each step of disassembly is cell-assisted, involving individual pathways that transmit signals to regulate discrete functions, but at the same time, these signalling pathways are organized into larger networks, which communicate back and forth in complex ways in response to the presence of virus. In this review, we consider the elegant strategy by which adenoviruses (AdVs) target and navigate cellular networks to initiate the production of progeny virions. There are many remarkable aspects about the AdV entry program; for example, the virus gains targeted control of a large well-defined local network neighbourhood by coupling several interacting processes (including endocytosis, autophagy and microtubule trafficking) around a collective reference state centred on the interactional topology and multifunctional nature of protein VI. Understanding the network targeting activity of protein VI, as well as other built-in mechanisms that allow AdV particles to be efficient at navigating the subsystems of the cell, can be used to improve viral vectors, but also has potential to be incorporated for use in entirely novel delivery systems.

## 1. Introduction

Viral particles at first glance may appear as little more than a genome enclosed in a simple protein cage; often times highly symmetrical in nature and in some instances wrapped in a lipid bilayer. Structural biology has dispelled this notion, especially as cryo-electron microscopy (cryo-EM) has matured into a method that can routinely determine near-atomic-resolution (2–4 Å) structures of viral assemblies [1]. The various architectural forms visualized by cryo-EM represent marvels of biological engineering so complex that even at high resolution it is often difficult to build precise and complete atomic models that accurately reflect virion organization (consider the recent spectacular work done on the herpesvirus capsid [2,3]). Even more challenging is the task of achieving a structural understanding of cell-mediated disassembly of virus particles from the metastable state to genome release [4].

The technical ability to correlate changes in viral structure with discrete steps in cell entry has largely been beyond reach, and even trying to simulate the process in a computer is too difficult due to the context, size and time scale of capsid uncoating. Nevertheless, researchers are making remarkable progress by shifting from using a single technology to embracing hybrid approaches that span different resolution scales [5,6]. In the case of adenoviruses (AdVs), the pay-off from experimental efforts has been significant. Research on the virus, as both a experimental model and as a target for intervention, has promoted the development of new methods (e.g. vitrification for cryo-EM [7]), taught us about fundamental cellular processes (e.g. RNA splicing [8]), and may one day play a role in the

prevention and treatment of human diseases (e.g. as vectors for vaccines and gene therapy [9]). Here, we discuss how AdV movement and disassembly occurs in time, through space, and by some kind of microenvironmental force, effort or energy, so that incoming viral genomes are released at the appropriate location for replication.

## 2. Cell networks are open but protective against viral infection

A typical mammalian cell contains tens of thousands of molecular components that are architecturally organized into networks to execute integrated functions such as growth, movement and communication [10]. Biological activity in the system is reversible, occurring mainly through manipulation of intermolecular noncovalent interactions, which enables diversity in cellular composition, either by internal reshuffling or by removal, exchange and incorporation of the various components (water, ions, organic molecules, carbohydrates, lipids, proteins and nucleic acids). Interactions are in part programmed, but can also be trained as the system is opened and coupled both spatially and temporally to the immediate surroundings. The capacity for programmability and adaptability in the different networks that comprise the molecular program partly explains why viruses have been a constant threat to cells for billions of years. However, host cell compartmentalization and the built-in molecular recognition features of networks make it challenging for viruses to enter and execute the production of new particles within the intracellular space [11].

Self-defence against infection begins with the organization of the superstructure of the cell [11]. Complex self-organization arises concomitantly during multimodal self-assembly, where the interactional recognition networks responsible for the formation of cellular structures only work if the system absorbs and dissipates energy from the environment's external drives. That is, stop the flow of energy and the cell dies. Organization is multimodal in the sense that it involves both equilibrium (e.g. globular protein folding, base pairing, lipid bilayer formation) and dynamic (e.g. actin filament formation, protein aggregation mechanisms, chromatin organization) self-assembly processes [12]. The result is a robust highly compartmentalized superstructure that restricts chemical events to spatially confined, functionally well-defined domains, so as to avoid internal build-up of disharmonic outputs that could diminish cell vitality, possibly altering either self-replication or survival capacities [13].

The first barrier to entry is the plasma membrane itself, which is heterogeneous by nature, featuring distinct subcompartments that differ in biophysical properties and composition [14]. Only small chemically suitable molecular species diffuse across the lipid bilayer either passively or by some input amount of energy, whereas for all other internalization events, surface receptors shift the dynamic equilibrium towards preferential formation of the best binding partners for further downstream processing. Cargo internalized via endocytic pathways are not free to roam but are fed directly into an uptake network rich in informational cues that couples vesicular transport to chemical potentials, energy, and signal transduction for rapid chemical transformation of incoming substrates [15]. This type of compartmentalized intracellular trafficking represents a second physical barrier to infiltration. The nuclear envelope is a third barrier that partitions the DNA of the cell away from the cytoplasm. Even more, it works like an integrated interface, providing a dynamic link between nuclear components and the cytoskeleton [16]. Small molecules less than approximately 40 nm may pass across the nuclear envelope unimpeded [17], but larger cargo must bear appropriate signals for transit across annular junctions via nuclear pore complexes [18,19].

Transport across the spatially embedded networks of the cell requires sharing of intracellular spaces and molecular components. Security is high priority and the cell has an impressive multilayered defence strategy mediated by effector molecules and compartment-specific sensory receptors that detect so-called pathogen-associated molecular patterns (PAMPs) and danger-associated molecular patterns (DAMPs). Incorporation of pattern recognition receptors (PRRs) into the diverse spaces of the superstructure of the cell, such as in the membranes or the cytoplasm, endows the cell with many 'informed' surfaces on which detection can occur [20]. These include Toll-like receptors, C-type lectin-like receptors, nucleotide binding and oligomerization domain-like receptors, cytoplasmic double-stranded DNA receptors and RIG-I-like receptors. The detected signal may be a cell-bound virus, a viral genome in the midst of a carrier process, or any molecule that gets released or secreted from damaged or dying cells [21–23]. Stimulation of a given sensory receptor triggers a cascade of tens or even hundreds of proinflammatory proteins that together orchestrate the early response to infection, and also play a role in activation, maturation and shaping of the adaptive immune response. In addition to the PAMP-recognition system, the cell can also detect host-derived antibody that gets internalized along with virus during cell entry [24]. Here, cytosolic antibody receptor tripartite motif protein 21 (TRIM21) binds to antibody-decorated AdV, specifically the Fc region of the antibody bound to the capsid, and then becomes activated to recruit ubiquitin-conjugating enyzmes (E2s: Ube2 W and Ube2n/Ube2v2), which build K63-linked ubiquitin chains onto TRIM21 [25]. This sends the signal that the virus is to be rapidly degraded by the proteasome, and at the same time K63-linked unbiquitin chains stimulate innate immune signalling pathways (NFKB, AP-1 and IRF3/5/7) to establish an antiviral state [26]. Following degradation of the viral capsid, exposure of the AdV DNA to the cytosolic DNA sensor cGAS triggers a second cascade of signalling events to further propagate the immune response [27]. Taken as a whole, the multilayered defence program of the cell provides the foundation for control and rescue of network dynamics, which is of paramount importance when faced with a virus that could potentially cause the entire system to change behaviour or fail.

## 3. Adenoviruses are built for cellular networking

Viruses thrive only to the extent that they are able to transmit their genomes from an infected host to a noninfected recipient. Once inside the new host, virions bind to permissive target cells followed by entry and replication, after which they can spread to neighbouring cells and repeat the process

royalsocietypublishing.org/journal/rsob    Open Biol. 9: 190012

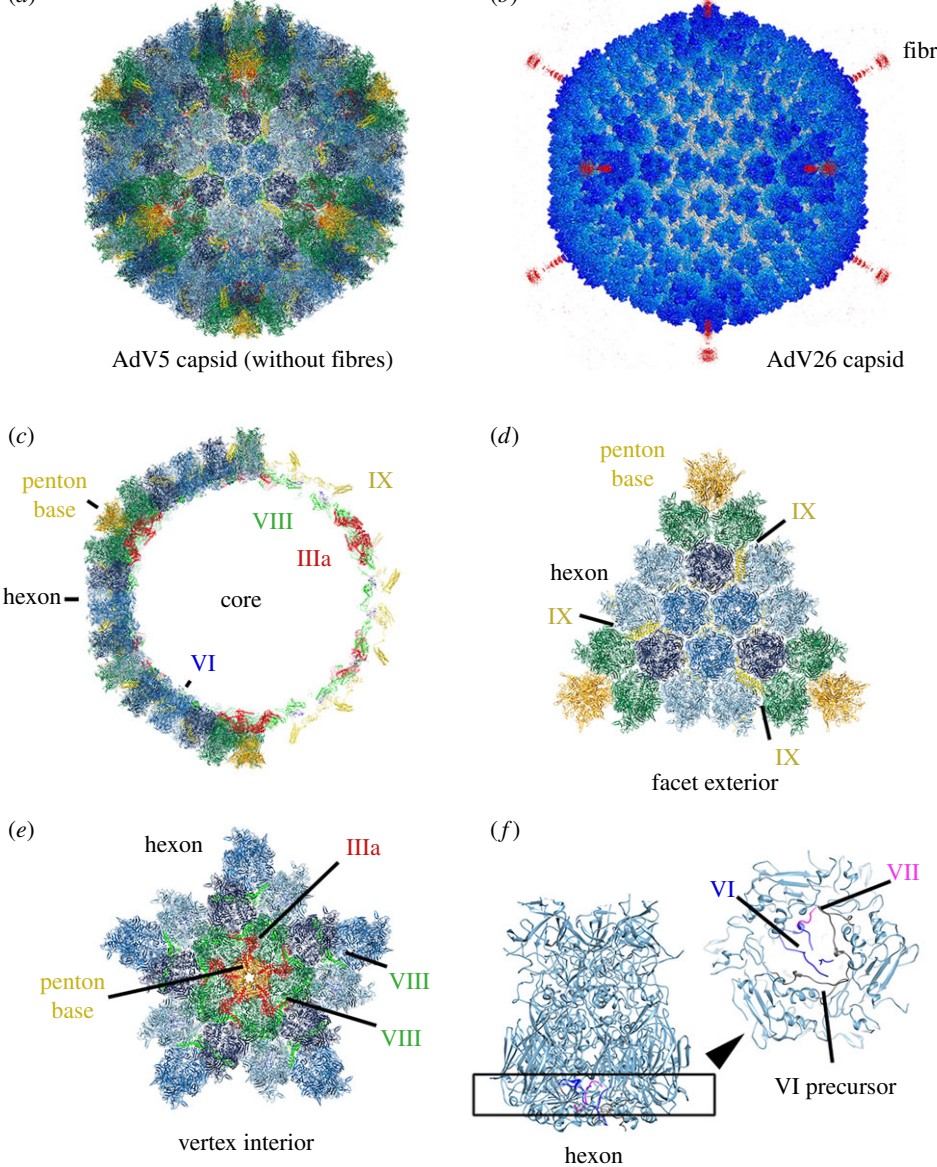

**Figure 1.** AdV capsid organization. (*a*) Capsid building blocks assemble to form approximately 900 Å icosahedrally symmetric shells. (*b*) AdV26 has a relatively short fibre that is visible by cryo-EM (accession code EMDB-8471). (*c*) The nanoscale container is defined by penton base and hexon proteins that occupy pentavalent and hexavalent positions on the pseudo *T* = 25 lattice, and the arrangement is stabilized and functionalized by layers of minor protein interactions (IIIa, VI, VIII and IX). (*d*) Minor protein IX is inlaid on the outer surface of the virion. (*e*) By contrast, minor proteins IIIa, VI and VIII occupy sites on the inner surface of the capsid. (*f*) Protein VI and the cleaved N terminus of protein VII bind to the inside of the hexon cavity.

[28]. Entry critically relies on a capsid, which not only serves as a protective shell that encloses the genome, but also contains built-in mechanisms that allow for coordinated intracellular movements and controlled release of the genome at the proper location within the cell. In this regard, AdVs are a robust family, composed of more than 100 known serotypes that can infect various vertebrate species including mammals, birds, fish, reptiles and amphibians [29]. To date, there are at least 57 serotypes divided into seven species (A–G) that can infect humans, causing acute respiratory disease, gastroenteritis, keratoconjunctivits and even obesity [30,31]. These diseases are generally self-limiting, but severe and deadly infections can occur in immunocompromised hosts [32]. Cell biology studies using optimized tissue culture systems have shown that typical times for producing AdV virions from infected cells ranges from 14 to 19 h depending on serotype [33], though the dynamics may differ *in vivo* given that these systems do not

fully capture all of the biological driving forces that exert selective pressures on the virus during entry.

The AdV capsid (not including the fibres) has an overall diameter of about 900 Å and is composed of major (hexon, penton base and fibre) and minor (IIIa, VI, VIII and IX) proteins [34] (figure 1*a*). These building blocks when assembled adopt an ancient underlying structure common to many viruses, the icosahedron: each of the 20 faces are composed of 12 copies of a trimer with pseudo-hexagonal character called hexon, and the vertices are formed by pentamers of penton base, each having one or more copies of a non-covalently associated trimeric spike referred to as a fibre (figure 1*b*). The fibres for a given AdV serotype vary in length depending on the number of β spiral repeats in the shaft [35]. Furthermore, some AdVs have more than one type of fibre incorporated into their virion, such as the Mexican beaded lizard AdV, designated lizard AdV2, which has either one short or three long fibres per penton

base [36]. The physical size of the fibre, along with sequence variation (hypervariable regions) in the three major capsid proteins, are major determinants of the entry pathway and immunological profile for the different serotypes [37].

In addition to hexon, penton base and fibre, there are four minor proteins intricately positioned in the virion that affect capsid assembly, disassembly and stability, which are often referred to as cementing or glue proteins (figure 1c). These proteins play important roles in cell entry and infection. The structures and organization of three of these minor proteins (IIIa, VIII and IX) has been a source of confusion and debate ever since the first atomic structures of intact AdV5 were independently derived by cryo-EM and X-ray crystallography [34,38]. Conflicting models were produced [39,40], but fortunately, the issue has been sorted out with agreement that protein IX is located on the outer surface of the virus particle, whereas proteins IIIa, VI and VIII are located on the capsid interior [41–43]. At the outside, 240 copies of protein IX are embedded between hexons, forming an external interaction network that stabilizes the icosahedral facets (figure 1d). Below the capsid surface at each of the 12 vertices, five copies of protein IIIa cement the gaps between each penton base and its five surrounding hexons. Also, IIIa molecules bind to adjacently positioned protein VIII, which has the effect of extending the cement to hexons beyond the peripentonal region (figure 1e). Multiple copies (120 per virion) of protein VIII glue neighbouring hexons together at threefold and fivefold sites on the capsid interior. Finally, protein VI can occupy up to three binding sites in the cavity of trimeric hexon (one site per subunit; 720 sites per virion), but must compete for attachment with core protein VII, and this explains why only approximately 350 copies of protein VI are found in the assembled virion (figure 1f).

During cell entry, environmental cues progressively induce conformational changes in the structural proteins of the incoming virion to promote stepwise disassembly, and to direct the virus to the site of replication. Once the uncoating program is completed, the virus is able to release the core that contains all the necessary information to initiate a productive infection, as well as to carry out synthesis and assembly of virus particles. The core itself consists of the genome, which is approximately 36 kb of double-stranded DNA, and is accompanied by core proteins V, VII, $\mu$, IVa2, terminal protein and viral protease. During assembly, this region of the virion condenses into a fluid-like unstructured state in the immature particle [44], which then undergoes proteolytic maturation via a protein VI-activated viral protease to produce mature infectious virions [45]. At this point, particles can be released from cells or from the infected host, and will withstand the stresses encountered in the extracellular space while en route to new permissive cellular environments. Capsids are sturdy enough to prevail against the internal pressure of the confined genome, which for AdVs is estimated to be approximately 30 atm [46], and at the same time, they can be used for directed transport and controlled release once inside cells [47,48].

## 4. Receptor binding and initiation of the uncoating program

The goal of AdV entry is to enter into the cell, bypass the security features and send the message for temporal programming of the production of virions. The overall process is instructed (target-driven) and dynamic, and combines both molecular and supramolecular events in such a way that the target with help from the cell ultimately performs assembly of the optimal parts to make new mature particles [49]. Furthermore, it involves adaptive chemistry in that the viral population may vary in composition not only by virtue of the properties of the different polymer components (e.g. mutations to the genome) but also because of the selection pressure exerted by changes in either the constituent parts or in the environmental conditions [50]. Such changes may give rise to performance enhancements that move towards generation of the fittest virions for replication, but this comes at a steep cost as viral production consumes host resources, damages host tissues, and provokes immune clearance, all of which shorten the infection period. Most of what we currently know about AdV entry pathways is the result of experimentation using either AdV2 or AdV5 at a high multiplicity of infection (MOI) in conventional cell culture systems (e.g. HeLa cells). This review is thus largely AdV2/5-centric; however, it is important to note that the virus type, cell type and host species significantly influence the entry pathway, as well as the associated immune response [51,52]. Furthermore, in the context of infection, one should keep in mind that when cells contain large numbers of virus particles, emergent properties arise that are best understood at the level of the system rather than by focusing on atom-by-atom interactions.

To begin the entry process, AdVs must first initiate contact at the plasma membrane of the target cell, and coordinate transport services to gain access to the internal networks (figure 2 receptor binding) [53]. This first recognition event is driven by the information stored in the shell-forming polymers of the virion. Permissive cells possess receptors that are activated on binding the virion, which is dictated by complementary matching based on size and shape, as well as the intrinsic energetic and stereochemical features of the noncovalent, intermolecular forces (hydrogen bonding, electrostatics, van der Waals, etc.) present at the virus-receptor interface. Each of the three major capsid proteins of the AdV virion bind to receptors on host cells. More specifically, the terminal knob domain of the AdV fibre is recognized by a variety of receptor molecules including the coxsackie and adenovirus receptor (CAR) [54–56], desmoglein-2 [57], CD46 [58–60] and sialic acid-containing glycans [61–64]. Loops of penton base that contain an Arg-Gly-Asp (RGD) motif bind to cellular integrins [65–67]. Hexon interacts with scavenger receptor SR-A6 [68,69], or with coagulation factors that then attach to heparan sulfate on hepatocytes [70–72]. The above list of binding partners is by no means exhaustive. Rather it represents the currently known receptors that display steric features complementary to components of the viral capsid alone or to virions in complex with host proteins. The best-studied example of a signal generating surface recognition event in the context of an AdV infection involves the selective binding of AdV2/5 by CAR and $a_v$ integrins. Here, attachment is a two-step process where different ligands on the virus particle are targeted by multiple receptors on the same cell. First, the higher-affinity receptor CAR displays recognition towards the knobs of the fibres protruding from the capsid, a step that anchors the virus to the cell. Second, $a_v$ integrins bind to penton base loops bearing RGD motifs. Impressive is the fact that the

royalsocietypublishing.org/journal/rsob   Open Biol. 9: 190012

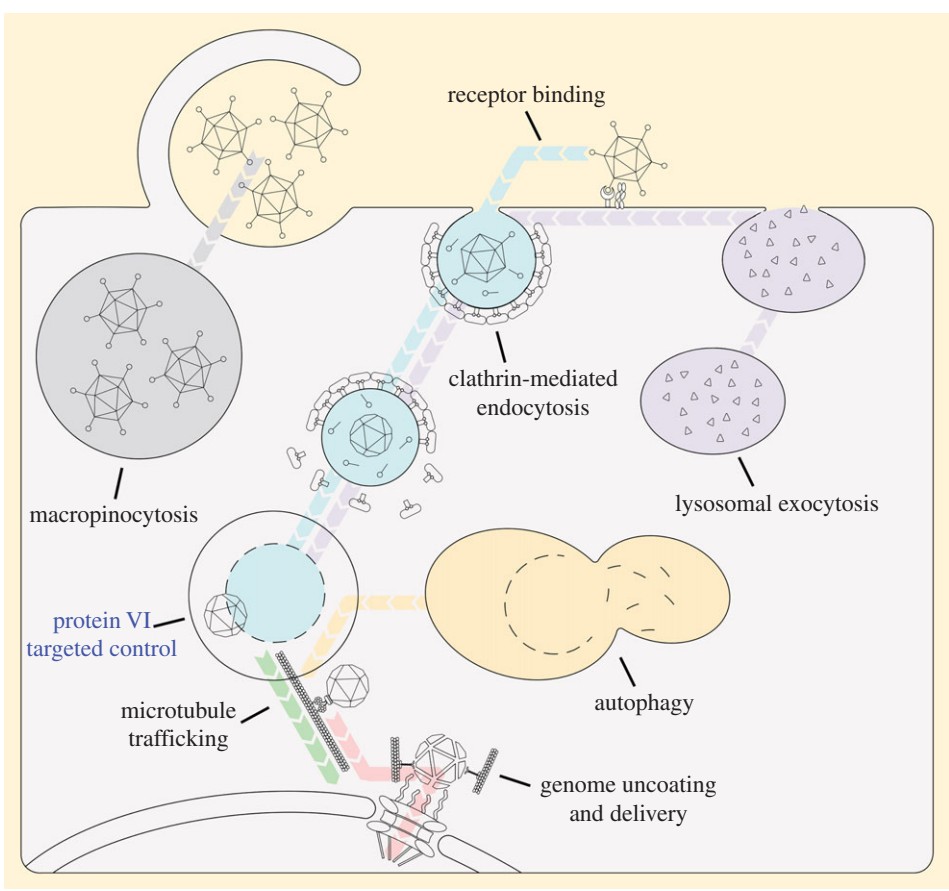

**Figure 2.** Target control of complex networks during AdV entry. AdVs make use of multiple intracellular pathways in parallel to deliver their genomes to the nucleus. At the point of endosomal escape, incoming virions utilize protein VI to gain control over a subset of nodes to establish the proviral state.

AdV fibre shaft flexes dramatically so that the virus can simultaneously engage both CAR and $a_v$ integrin receptors to form the tethered ligand assembly [73].

At the point of surface attachment, the metastable static viral particle reaches a committal step by entering a dynamic state, subject to motional and constitutional dynamics (uncoating), which proceeds under the control of cellular cues in a stepwise, irreversible manner as virus particles move through the cell until the eventual release of the genome at the site of replication [74]. The coupling of CAR and integrin into the tethered assembly introduces both global and local non-uniform motions to the virus particle. That is, CAR-engaged AdV diffuses and undergoes actomyosin dependent drifts on the surface of the cell for many seconds, after which motion becomes more confined along the axis of the virus fibre as penton base subunits interact with $a_v$ integrins [75]. This type of receptor engagement produces triggered molecular motions (local deformations, translations and rotations) and quite possibly conformational changes that lead to release of components (fibre and protein VI) from the AdV capsid [76,77]. Thus, the binding of AdV by CAR and $a_v$ integrins serves the dual role of generating a first uncoating intermediate and signalling uptake into the endocytic network.

## 5. Endocytic uptake

Activation of $a_v$ integrins by AdV penton base RGD loops triggers the transient formation of endocytic machinery on the plasma membrane to package the virus into a clathrin-coated vesicle for cellular internalization (figure 2 clathrin-mediated endocytosis) [78]. Site-directed assembly is initiated on the cytoplasmic leaflet of the plasma membrane and is guided by discrete spatio-temporally controlled binding events involving more than 50 cellular proteins organized into distinct functional modules. These modules form extensive intra- and interspecific contacts, of which formation is coupled either directly or indirectly to signalling molecules, including enzymes that control phospholipid metabolism, membrane remodelling, phosphorylation/dephosphorylation and cytoskeletal architecture [79]. A key benefit to the modular system architecture is the reversible nature of the non-covalently linked building blocks, which allows the cell to reuse the system and its components. The endocytic process is initiated at the site of the bound virus, which results in recruitment of adaptor proteins such as AP2 and EPS15 to provide additional docking sites for scaffold proteins and clathrin [80]. The assembling coat, by virtue of being rigid, deforms the plasma membrane to form a small approximately 100–200 nm virus-loaded pit. Other drivers of deformation include localized actin dynamics, membrane tension, the size and symmetry of the AdV particle, lipid composition and utilization of auxiliary proteins that contain domains or motifs specialized in sensing, creating and/or stabilizing membrane curvature [81]. Conversion of the clathrin-coated pit into a spherical carrier requires a scission step to separate the vesicle from its parent membrane. Here, the GTPase dynamin assembles into a collar around the neck of the budding coated pit and uses a GTP hydrolysis reaction to release the vesicle inside the cell [82]. Actin polymerization plays an essential role in

the scission mechanism. The newly formed vesicle undergoes rapid clathrin-lattice disassembly, a process mediated by the chaperone Hsc70 and cochaperone auxilin [83], so that the virus is in a membrane-enclosed compartment for intracellular trafficking.

An alternative route used by AdV to enter cells relies on subversion of an endocytic process referred to as macropinocytosis, which involves the transient nonspecific uptake of large fluid droplets (figure 2 macropinocytosis) [84]. Delivery via macropinocytic function is triggered by $a_v$ integrin-bound virus and coincides with clathrin-mediated endocytosis [85]. It is accompanied by dramatic rearrangements of filamentous actin, which causes cell-wide plasma membrane protrusions and ruffling. In the case of AdV, formation of large, motile, flattened lamellopodia protrusions supports formation of fluid-filled virus-containing cavities that can undergo membrane fission. Emerging macropinosomes share striking similarities to endosomes; they move to deeper locations in the cell, show sensitivity to cytoplasmic pH, become increasingly acidic, and participate in homo- and heterotypic fusion events. The non-specificity inherent in the uptake of fluid via macropinocytosis is thought to be advantageous to viruses as it relaxes the constraints on internationalization by bypassing the need for bound virus to associate with an entry-activating receptor. However, for AdV the picture is very complex in that there appears to be a close relationship between the parallel functioning macropinocytosis and clathrin-mediated endocytosis pathways. Evidence suggests that AdV endosomal escape is dependent on having an activated macropinocytic pathway, though such a connection may only be relevant at high MOI [85]. This type of cross-talk in the membrane-delimited entry and cargo sorting routes of the endocytic network is likely a manifestation of an emergent property where unexpected behaviours stem from interaction between the components and layers of the two uptake subsystems in the presence of virus.

## 6. Rupturing the endocytic network

Top-down analysis of virus flow during cell-wide endocytosis has led to a rich understanding of the link between capsid uncoating and temporal tuning of cellular parameters to promote trafficking far beyond the plasma membrane. The endocytic system is a dynamic and robust network through which extracellular inputs can be internalized by the cell in membrane-enclosed vesicles and then transported to their intended destination, typically either directed for degradation via lysosomes or marked for recycling back to the plasma membrane. The order of events is a contextual attribute of strict and quantifiable patterns of interactions between many cellular constituents (vesicles, effectors, cytoskeletal motors, organelles), which the virus can manipulate using its built-in uncoating program. Uncoating is a triggered dynamic process that results in the production of important constitutional changes: fluctuations, conformational alterations and releases of mechanically maintained components of the assembled virus. The strategy depends on both the cell and the viral capsid, with the latter relaying cues from the environment to induce controlled structural modifications [86]. In the case of AdV, the uncoating program ensures timely release of a multifunctional inner capsid protein that has membrane lytic properties, protein VI, which is capable

of rupturing flows on the endocytic network before the virus can be directed to the lysosome for degradation [87–89]. The rupturing capacity of protein VI, as well as its role in regulating capsid stability and endosomal escape, have been studied in great detail using a temperature sensitive mutant of human adenovirus 2 and a mutant virus with single point mutation in in protein VI, L40Q [89–93].

The mechanism underlying virus rupture of the endosome exploits the interconnection regularity inherent in the uptake system such that controlled build-up of protein VI occurs through a defined sequence of releases as the capsid disassembles during intracellular trafficking. The first release of protein VI occurs when the virus interacts with receptor molecules on the surface of the target cell. Here, CAR-engaged fibre, coupled with $a_v$ integrin binding to penton base, generates excessive disruptive motions strong enough to cause a vertex-specific tearing event, which results in the formation of a fibreless leaky capsid at the outset of endocytosis. Exposure of protein VI at the site of uptake on the surface of the cell causes membrane piercing, which activates another cellular network, namely a lysosomal exocytosis repair pathway that works to maintain cellular membrane integrity (figure 2 lysosomal exocytosis) [94]. The network is activated as protein VI-mediated lesions bleed calcium into the cell, and results in delivery of acid sphingomyelinase to the outer surface of the plasma membrane. At the surface, acid sphingomyelinase generates ceramide through the hydrolytic removal of the phosphorylcholine head group of sphingomyelin to promote inward bending and budding of membranes in a special type of injury-dependent endocytosis operation aimed at repairing the virus-inflicted wound. At the same time, the ceramide-enriched membrane accelerates viral endocytosis and enhances the bilayer-destabilizing interactions of protein VI. The second release of membrane lytic protein VI occurs inside the early endosome as vertex regions are removed from the AdV capsid, and at this point the endocytic vesicle breaks [91,95]. In this way, cross-talk between clathrin-mediated endocytosis and lysosomal exocytosis networks gives rise to collective dynamics that favour fast-tracked uptake.

## 7. Escape from endosomes and host detection

Breaking the endocytic network is a major development in AdV entry. The virus has managed to usurp the uptake system, something the cell senses as dangerous. In particular, both virus attachment and endosomal membrane rupture represent threat-specific cues that trigger immune responses to prevent flows of AdV within the cell [96,97]. What ensues is a perpetual struggle for existence where despite the cell's best efforts to trap infections, the virus may succeed by relying on stepwise disassembly of its capsid to gain progressive control of the situation. An example of capsid-based circumvention occurs immediately after endosomal rupture, an event which releases a recognizable pattern of physiological network dysfunction. Specifically, during endosomal breakage and virus escape, which are spatially and temporally distinct events [95], compartmentalized glycans are released to the cytosol, where in the foreign context they elicit an autophagic response that is selective in eliminating the damaged components that could become toxic to the

royalsocietypublishing.org/journal/rsob    Open Biol. **9**: 190012

royalsocietypublishing.org/journal/rsob     Open Biol. **9**: 190012

cell (figure 2 autophagy) [98]. When this happens, partially disassembled AdV is able to avoid autophagic degradation by use of a PPXY motif present in protein VI, which recruits the E3 ubiquitin ligase Nedd4.2 away from its role in regulating formation of autolysosomes [99]. Furthermore, AdV rewires autophagy to enhance endosomal escape and accelerate transport towards the microtubule-organizing center (MTOC) via unknown mechanisms. And finally, protein VI PPXY-mediated manipulation and avoidance of the autophagic response has immune implications as autophagy is connected to the antigen presenting network, which is critical for effective adaptive immunity [99,100]. Intriguingly, Paneth cells in the small intestines prevent the network from collapsing into a proviral configuration by expressing effector peptides termed α-defensins that target incoming capsids to block uncoating and release of protein VI during entry [101,102]. In such instances, defensin-coated virus particles do not rupture endosomes but are escorted to lysosomes for degradation [103].

## 8. Cytoplasmic transport

Emergence of virus within the cell is marked by a shift in traffic dynamics, during which there is a transition from isolated local flow inside the endosomal compartment to global flow in the cytoplasm. Transport from this moment forward requires that AdV integrate into the global flow of the cell, which is difficult given the complex spatial and temporal features of the transportation infrastructure and the high degree of mobility fluxes and heavy molecular traffic. To move with the flow, the virus mimics the behaviour of the intracellular crowd in relying on ATP-fuelled molecular motors for achieving directed movements along cytoskeletal network of filamentous proteins [104]. Specifically, AdV uses the architecture and nucleotide-dependent conformational changes of the dynein motor and its cofactor dynactin for long-distance minus-end-directed movements along microtubules (figure 2 microtubule trafficking) [105–109]. Exactly how incoming virus particles merge into cell-wide traffic is incompletely understood, but it appears that an uncoating-specified molecular conformation of hexon interacts directly with phosphorylated dynein to actuate the process [110]. Also, AdV exploits cell communication mechanisms such as p38 MAPK family and protein kinase A signalling to ensure rapid access to a stable microtubule track [111]. The virus-occupied motor-driven complex moves coherently in discrete steps (changes of velocity, transient pauses and direction reversals) until reaching the MTOC close to the nucleus. Offloading at the MTOC is supported by a perinuclear gradient of nuclear export factor CRM1, which tunes movements to a slower pace and regulates binding of the motor to its cargo [112,113]. Thus, the transition from isolated local flow in the endosome to congested global flow in the cytoskeletal network is influenced by many factors including motor recruitment/coordination, microtubule architecture and dynamics, traffic volume, and spatial cues from the local microenvironment. Ultimately, the polarity of dynein–microtubule interactions and the regulatory activity of CRM1 at the MTOC create a bias in the direction of motion, allowing AdV to target the nucleus for genome release.

## 9. Genome uncoating and delivery at the nuclear pore complex

For AdV particles that successfully navigate the superstructure of the cell to reach the nucleus, the last major barrier is the nuclear envelope (NE). The NE serves not only as a physical bilayer partition, but also as an information-processing centre for regulated exchange of matter between the nucleus and cytoplasm. The latter function is made possible by incorporation of dynamic channel-forming nuclear pore complexes (NPCs), each consisting of hundreds of proteins called nucleoporins (Nups), which include both structural Nups that build the scaffolding structure of the pore and phenylalanine-glycine-rich (FG-rich) intrinsically disordered Nups that emanate as fibrils into the cytoplasm and line the central channel to provide a selective passageway for receptor-mediated active transport [114,115]. Large macromolecules are granted passage through the NPC by interacting with transport factors that are capable of shuttling cargo past the otherwise impenetrable FG filter. AdV is an unusual cargo in that it first docks to the NPC via surface-to-surface interaction between the partially disassembled hexon shell and Nup214 [116,117], and then undergoes a final uncoating step to release the core of DNA for import through the central channel (figure 2, genome uncoating and delivery) [118]. The capsid dismantling event is a tug-of-war-mediated process in which the Nup214-anchored virus recruits kinesin-1 motors to protein IX molecules at the surface of the bound virus particle [119]. Afterwards, the NPC itself (Nup358) activates the recruited kinesin motors for movement along proximal microtubules, which gives rise to multidirectional mechanical tugs sufficient to uncoat the viral genome for delivery. Once released, the double-stranded DNA is associated with hundreds of copies of nuclear localization signal (NLS)-containing protein VII molecules that bind to nuclear transport receptors to help facilitate rapid transport through the NPC [120]. However, the stress of uncoating and subsequent genome delivery disrupts NPC architecture [119] and can lead to misdelivery of viral DNA to the cytosol [121,122]. Also, during the final stage of delivery, protein VI molecules mediate import of newly synthesized hexon into the nuclear compartment so that virus assembly can occur. Protein VI is able to shuttle between the cytoplasm and nucleus because it contains nuclear import and export signals in its C-terminus [123]. In the cytoplasm, protein VI forms a complex with hexon, and then recruits importins α and β to promote translocation through the NPC [123]. Following formation of new capsids, maturation results in removal of the C-terminal transport signals from protein VI, which switches the functionality of protein VI away from supporting hexon import to a structural role in virus assembly [123].

## 10. Conclusion

In summary, transport in complex networks is a problem of much interest in many aspects of biology. In this regard, the study of virus entry has been crucial for gaining a deeper understanding of how cells regulate and coordinate various events across their internal networks. Also, viruses provide examples of how to achieve targeted control of intracellular subsystems, which may be interesting for a variety of

applications. Here, we have looked at how AdVs navigate the connected pathways to deliver their genomes for production of progeny virions with particular emphasis placed on the multifaceted role of viral protein VI during the entry process. Protein VI contains an N-terminal amphipathic helix that fragments the endosomal membrane and an adjacent PPXY motif that is exposed upon membrane lysis to help escape the endocytic network and tune autophagy and microtubule trafficking to a proviral state. The target control efficiency of protein VI is impressive given that intracellular networks display a high degree of robustness, an attribute due in part to the redundant wiring of functions within the integrated system as a whole. Local failures rarely result in the loss of normal functional states; however, as research on protein VI has demonstrated, error tolerance comes at a high price in that the system is extremely vulnerable to attacks, especially when the target is a few critical nodes that play a key role in maintaining the network's connectivity. Future work is needed to help clarify the network targeting activity of protein VI, as well as other built-in mechanisms that allow AdV particles to be efficient at navigating the intracellular space for temporal programming of the production of virions. In undertaking such work, we will learn more about how cells and viruses work, and at the same time, the knowledge can be used to improve vectors for basic research and therapeutic applications, either by optimizing AdVs for delivery or by transferring certain properties (like protein VI targeted control capacity) to other types of delivery systems.

Data accessibility. This article has no additional data.

Competing interests. We declare we have no competing interests.

Funding. We would like to acknowledge funding from the Sigrid Juselius Foundation (SJB), the Academy of Finland (315950) and Helsinki Institute of Life Sciences.

Acknowledgements. We thank Dr Maria Anastasina (University of Helsinki) for help with making figure 2.

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
