## [Reviewer comments · Open Biology]

Review History

RSOB-19-0012.R0 (Original submission)

Review form: Reviewer 1

Recommendation

Accept with minor revision (please list in comments)

Are each of the following suitable for general readers?

- a) **Title**
Yes
- b) **Summary**
Yes
- c) **Introduction**
Yes

Is the length of the paper justified?

Yes

Should the paper be seen by a specialist statistical reviewer?

No

Is it clear how to make all supporting data available?

Not Applicable

Is the supplementary material necessary; and if so is it adequate and clear?

Not Applicable

Do you have any ethical concerns with this paper?

No

Comments to the Author

Comments on MSRSOB-19-0012 Open Biology

Authored by: Justin Flatt & Sarah Butcher

This review summarizes the events occurring during adenovirus entry from cell attachment to genome delivery at the nuclear pore. The authors focus their attention on the role of the internal capsid protein VI as a driving and coordinating force of adenovirus entry. In addition, the authors present the cell as an obligatory open functional network that is susceptible to viral intervention secured by intrinsic control mechanisms.

On the whole, this manuscript is well written largely covering the existing literature. The authors favor an unorthodox writing style, which in my opinion sometimes lacks immediate clarity (see below). Still, I enjoyed reading this review and think it provides a comprehensive and interesting read.

In principle, the authors provide a description of generic adenovirus entry, which is legitimate, often referring to references using in depth characterization of HAdV-C2 or C5 at high MOI in conventional cell culture models (e.g. HeLa cells). An important point that the authors should mention more clearly is that host species, cell type and virus type can substantially influence the entry pathway and the associated immune response (e.g. PMID:17707546, PMID:11152512). E.g. most of the TRIM21 work was performed in murine cells, while AdV is not permissive in murine cells.

The authors may consider the following points

- Protein VI is certainly not the only protein directing entry even if it has been investigated more profoundly and plays an important role. Thus, stating that "AdV gains targeted control of the cell network with JUST A SINGLE protein" (abstract) should be put into context. Then again, protein VI plays an important role in assembly, maturation (protease activation), capsid stability (see below) and nuclear import of hexon, which is maybe worth mentioning.
- AdV (partial) disassembly during entry occurs at defined steps/barriers that allow to overcome the obstacle to continue with the entry process. Please avoid phrases that suggest otherwise (e.g. "As the virus travels through the cell ... cues progressively induce conformational changes (part 2), "particles can fall apart in the cell" (part 2).
- Part 2 is numbered twice and so is part 6
- Please clarify (simplify) the text in the cell networks part (part 2) especially the second paragraph. To have a more practical connection you may consider to combine this part with the description of the innate sensing at the end of the cytosolic transport part (part 7).

- Immune detection does not only result in autophagy and immunophagy (last sentence part 2)
- I slightly disagree with the term “introduction of foreign matter” in part 2. Is it not more an “out-of-place detection” (e.g. DNA in the cytosol or endosome, glycosylated proteins in the cytosol) ? (part2)
- Part 3 is very general and could be combined with part 4
- Point out that host species, cell type and virus type can substantially influence the entry pathway
- Macropinocytosis may only be relevant at high MOI. (part 5)
- The authors should consider to give the adenovirus temperature sensitive mutant AdVts1 and/or the AdV-pVI-L40Q mutant as an example for the importance of protein VI in regulating capsid stability, endosome penetration and endosomal escape (e.g. PMID:21209115, PMID:24027314, PMID:25473051, PMID:19563809) (part 5-6)
- Maybe it is worthwhile to point out more clearly that endosomal rupture and escape from the endosomal compartment are two separate events (e.g. PMID:22855481) (part 5).

Review form: Reviewer 2

Recommendation

Accept with minor revision (please list in comments)

Are each of the following suitable for general readers?

- a) **Title**
Yes
- b) **Summary**
Yes
- c) **Introduction**
Yes

Is the length of the paper justified?

Yes

Should the paper be seen by a specialist statistical reviewer?

No

Is it clear how to make all supporting data available?

Not Applicable

Is the supplementary material necessary; and if so is it adequate and clear?

Not Applicable

Do you have any ethical concerns with this paper?

No

Comments to the Author

This review article by Flatt and Butcher describes an overview and recent knowledge about cytoplasmic transport of adenoviruses, as well as interactions with host mechanisms during the process. By referring to structural and biochemical evidence, the manuscript widely covers topics,

from entry into host cells to genome uncoating at nuclear pore complexes, and is described in a well-balanced manner. Overall, I feel the manuscript provides the general, up-to-date view for adenoviral trafficking mechanisms and is of good quality as a review article. Therefore, I have only a few minor points that may need to be addressed as follows.

1. In the Abstract and Conclusions, protein VI appears to be featured among other viral proteins. However, the main text describes general topics on adenoviral transport without specific focus on the protein. Although it is clear that protein VI is a key player in several processes mentioned in the manuscript, the Abstract would need to be more balanced according to the contents of the manuscript.
2. I feel some parts of the text are a bit difficult to follow. For instance, page 1 gives a general introduction, page 2 explains specific structural features of adenoviruses, page 3 goes back to a general topic on host-virus interactions, and then page 4 is again about adenoviruses. Reorganizing the text would increase readability.
3. Subheads are mislabeled: 1. Introduction, 2. Adenoviruses are built for cellular networking, and "2". Cell networks are open but protective against viral infection.

Decision letter (RSOB-19-0012.R0)

04-Feb-2019

Dear Dr Flatt,

We are pleased to inform you that your manuscript RSOB-19-0012 entitled "Adenovirus flow in host cell networks" has been accepted by the Editor for publication in Open Biology. The reviewer(s) have recommended publication, but also suggest some minor revisions to your manuscript. Therefore, we invite you to respond to the reviewer(s)' comments and revise your manuscript.

Please submit the revised version of your manuscript within 14 days. If you do not think you will be able to meet this date please let us know immediately and we can extend this deadline for you.

- 1) A text file of the manuscript (doc, txt, rtf or tex), including the references, tables (including captions) and figure captions. Please remove any tracked changes from the text before submission. PDF files are not an accepted format for the "Main Document".
- 2) A separate electronic file of each figure (tiff, EPS or print-quality PDF preferred). The format should be produced directly from original creation package, or original software format. Please note that PowerPoint files are not accepted.
- 3) Electronic supplementary material: this should be contained in a separate file from the main text and meet our ESM criteria (see <http://royalsocietypublishing.org/instructions-authors#question5>). All supplementary materials accompanying an accepted article will be treated as in their final form. They will be published alongside the paper on the journal website and posted on the online figshare repository. Files on figshare will be made available approximately one week before the accompanying article so that the supplementary material can be attributed a unique DOI.

Online supplementary material will also carry the title and description provided during submission, so please ensure these are accurate and informative. Note that the Royal Society will not edit or typeset supplementary material and it will be hosted as provided. Please ensure that the supplementary material includes the paper details (authors, title, journal name, article DOI). Your article DOI will be 10.1098/rsob.2016[last 4 digits of e.g. 10.1098/rsob.20160049].

- 4) A media summary: a short non-technical summary (up to 100 words) of the key findings/importance of your manuscript. Please try to write in simple English, avoid jargon, explain the importance of the topic, outline the main implications and describe why this topic is newsworthy.

Images

Data-Sharing

It is a condition of publication that data supporting your paper are made available. Data should be made available either in the electronic supplementary material or through an appropriate repository. Details of how to access data should be included in your paper. Please see <http://royalsocietypublishing.org/site/authors/policy.xhtml#question6> for more details.

Sincerely,

The Open Biology Team
<mailto:openbiology@royalsociety.org>

Reviewer(s)' Comments to Author:

Referee: 1

Comments to the Author(s)
Comments on MSRSOB-19-0012 Open Biology
Authored by: Justin Flatt & Sarah Butcher

This review summarizes the events occurring during adenovirus entry from cell attachment to genome delivery at the nuclear pore. The authors focus their attention on the role of the internal capsid protein VI as a driving and coordinating force of adenovirus entry. In addition, the authors present the cell as an obligatory open functional network that is susceptible to viral intervention secured by intrinsic control mechanisms.

On the whole, this manuscript is well written largely covering the existing literature. The authors favor an unorthodox writing style, which in my opinion sometimes lacks immediate clarity (see below). Still, I enjoyed reading this review and think it provides a comprehensive and interesting read.

In principle, the authors provide a description of generic adenovirus entry, which is legitimate, often referring to references using in depth characterization of HAdV-C2 or C5 at high MOI in conventional cell culture models (e.g. HeLa cells). An important point that the authors should mention more clearly is that host species, cell type and virus type can substantially influence the entry pathway and the associated immune response (e.g. PMID:17707546, PMID:11152512). E.g. most of the TRIM21 work was performed in murine cells, while AdV is not permissive in murine cells.

The authors may consider the following points

- Protein VI is certainly not the only protein directing entry even if it has been investigated more profoundly and plays an important role. Thus, stating that "AdV gains targeted control of the cell network with JUST A SINGLE protein" (abstract) should be put into context. Then again, protein VI plays an important role in assembly, maturation (protease activation), capsid stability (see below) and nuclear import of hexon, which is maybe worth mentioning.
- AdV (partial) disassembly during entry occurs at defined steps/barriers that allow to overcome the obstacle to continue with the entry process. Please avoid phrases that suggest otherwise (e.g. "As the virus travels through the cell ... cues progressively induce conformational changes (part 2), "particles can fall apart in the cell" (part 2).
- Part 2 is numbered twice and so is part 6
- Please clarify (simplify) the text in the cell networks part (part 2) especially the second paragraph. To have a more practical connection you may consider to combine this part with the description of the innate sensing at the end of the cytosolic transport part (part 7).
- Immune detection does not only result in autophagy and immunophagy (last sentence part 2)
- I slightly disagree with the term "introduction of foreign matter" in part 2. Is it not more an "out-of-place detection" (e.g. DNA in the cytosol or endosome, glycosylated proteins in the cytosol) ? (part2)
- Part 3 is very general and could be combined with part 4
- Point out that host species, cell type and virus type can substantially influence the entry pathway
- Macropinocytosis may only be relevant at high MOI. (part 5)
- The authors should consider to give the adenovirus temperature sensitive mutant AdVts1 and/or the AdV-pVI-L40Q mutant as an example for the importance of protein VI in regulating capsid stability, endosome penetration and endosomal escape (e.g. PMID:21209115, PMID:24027314, PMID:25473051, PMID:19563809) (part 5-6)
- Maybe it is worthwhile to point out more clearly that endosomal rupture and escape from the endosomal compartment are two separate events (e.g. PMID:22855481) (part 5).

Referee: 2

Comments to the Author(s)

This review article by Flatt and Butcher describes an overview and recent knowledge about cytoplasmic transport of adenoviruses, as well as interactions with host mechanisms during the process. By referring to structural and biochemical evidence, the manuscript widely covers topics, from entry into host cells to genome uncoating at nuclear pore complexes, and is described in a well-balanced manner. Overall, I feel the manuscript provides the general, up-to-date view for adenoviral trafficking mechanisms and is of good quality as a review article. Therefore, I have only a few minor points that may need to be addressed as follows.

1. In the Abstract and Conclusions, protein VI appears to be featured among other viral proteins. However, the main text describes general topics on adenoviral transport without specific focus on the protein. Although it is clear that protein VI is a key player in several processes mentioned in the manuscript, the Abstract would need to be more balanced according to the contents of the manuscript.
2. I feel some parts of the text are a bit difficult to follow. For instance, page 1 gives a general introduction, page 2 explains specific structural features of adenoviruses, page 3 goes back to a general topic on host-virus interactions, and then page 4 is again about adenoviruses. Reorganizing the text would increase readability.
3. Subheads are mislabeled: 1. Introduction, 2. Adenoviruses are built for cellular networking, and "2". Cell networks are open but protective against viral infection.

Author's Response to Decision Letter for (RSOB-19-0012.R0)

See Appendix A.

Decision letter (RSOB-19-0012.R1)

07-Feb-2019

Dear Dr Flatt,

We are pleased to inform you that your manuscript entitled "Adenovirus flow in host cell networks" has been accepted by the Editor for publication in Open Biology.

Article processing charge

Please note that the article processing charge is immediately payable. A separate email will be sent out shortly to confirm the charge due. The preferred payment method is by credit card; however, other payment options are available.

Sincerely,

The Open Biology Team
mailto: openbiology@royalsociety.org

Appendix A

Dear Reviewers,

We thank you for reviewing the manuscript and providing useful feedback. We believe that the revised version has been greatly improved. Please find our responses to specific comments below.

Responses to Reviewer 1

In principle, the authors provide a description of generic adenovirus entry, which is legitimate, often referring to references using in depth characterization of HAdV-C2 or C5 at high MOI in conventional cell culture models (e.g. HeLa cells). An important point that the authors should mention more clearly is that host species, cell type and virus type can substantially influence the entry pathway and the associated immune response (e.g. PMID:17707546, PMID:11152512). E.g. most of the TRIM21 work was performed in murine cells, while AdV is not permissive in murine cells.

Thank you for the comment. We have added a couple of sentences and references to the manuscript (see section 4: Adenovirus delivery across cellular networks) to briefly discuss this important point.

In section 4:

"Most of what we currently know about AdV entry pathways is the result of experimentation using either AdV2 or AdV5 at a high multiplicity of infection (MOI) in conventional cell culture systems (e.g., HeLa cells). This review is thus largely AdV2/5-centric, however, it is important to note that the virus type, cell type, and host species significantly influence the entry pathway, as well as the associated immune response (45, 46)."

-Protein VI is certainly not the only protein directing entry even if it has been investigated more profoundly and plays an important role. Thus, stating that "AdV gains targeted control of the cell network with JUST A SINGLE protein" (abstract) should be put into context. Then again, protein VI plays an important role in assembly, maturation (protease activation), capsid stability (see below) and nuclear import of hexon, which is maybe worth mentioning.

The comments about protein VI are well taken. The text has been modified in several places.

In the abstract

Instead of "...topology and multifunctional nature of just a single viral protein, protein VI." it now reads "state centered on the interactional topology and multifunctional nature of protein VI."

In the last paragraph of section 3:

"...maturation via the viral protease..." is written "...maturation via a protein VI-activated viral protease..."

In section 9:

"Also, during the final stage of delivery, protein VI molecules mediate import of newly synthesized hexon into the nuclear compartment so that virus assembly can occur. Protein VI is able to shuttle between the cytoplasm and nucleus because it contains nuclear import and export signals in its C-terminus (119). In the cytoplasm, protein VI forms a complex with hexon, and then recruits importins α and β to promote translocation through the NPC (119). Following formation of new capsids, maturation results in removal of the C-terminal transport signals from protein VI, which switches the functionality of protein VI away from supporting hexon import to a structural role in virus assembly (119)."

-AdV (partial) disassembly during entry occurs at defined steps/barriers that allow to overcome the obstacle to continue with the entry process. Please avoid phrases that suggest otherwise (e.g. "As the virus travels through the cell ... cues progressively induce conformational changes (part 2), "particles can fall apart in the cell" (part 2).

In section 3 it now reads:

"During cell entry, environmental cues progressively induce conformational changes in the structural proteins of the incoming virion to promote stepwise disassembly, and to direct the virus to the site of replication."

and

"Capsids are sturdy enough to prevail against the internal pressure of the confined genome, which for AdVs is estimated to be ~30 atm (28), and at the same time, they can be utilized for directed transport and controlled release once inside cells (29, 30)."

-Part 2 is numbered twice and so is part 6

The numbering issue has been fixed.

-Please clarify (simplify) the text in the cell networks part (part 2) especially the second paragraph. To have a more practical connection you may consider to combine this part with the description of the innate sensing at the end of the cytosolic transport part (part 7).

Thank you for the suggestion to move the description on innate sensing into the section covering cell networks. We have reorganized the text (see Section 2: Cell networks are open but protective against viral infection) and this has nicely improved the flow of the manuscript.

-Immune detection does not only result in autophagy and immunophagy (last sentence part 2)

The sentence commenting about immune detection, autophagy, and immunophagy is no longer in the manuscript. It was removed when reworking the description of innate immunity into the section covering cell networks. See section 2. Cell networks are open but protective against viral infection in the revised version of the paper.

-I slightly disagree with the term "introduction of foreign matter" in part 2. Is it not more an "out-of-place detection" (e.g. DNA in the cytosol or endosome, glycosylated proteins in the cytosol) ? (part2)

The sentence containing "introduction of foreign matter" is no longer in the manuscript. It was removed when reworking the description of innate immunity into the section covering cell networks. See section 2. Cell networks are open but protective against viral infection in the revised version of the paper.

-Part 3 is very general and could be combined with part 4

We have now combined sections 3 and 4 (Section 4. Receptor binding and initiation of the uncoating program).

-Point out that host species, cell type and virus type can substantially influence the entry pathway

We have added text and references to section 4:

"Most of what we currently know about AdV entry pathways is the result of experimentation using either AdV2 or AdV5 at a high multiplicity of infection (MOI) in conventional cell culture systems (e.g., HeLa cells). This review is thus largely AdV2/5-centric, however, it is important to note that the virus type, cell type,

and host species significantly influence the entry pathway, as well as the associated immune response (45, 46)."

-Macropinocytosis may only be relevant at high MOI. (part 5)

We now mention in the section on macropinocytosis:

"Evidence suggests that AdV endosomal escape is dependent on having an activated macropinocytic pathway, though such a connection may only be relevant at high MOI (79)."

-The authors should consider to give the adenovirus temperature sensitive mutant AdVts1 and/or the AdV-pVI-L40Q mutant as an example for the importance of protein VI in regulating capsid stability, endosome penetration and endosomal escape (e.g. PMID:21209115, PMID:24027314, PMID:25473051, PMID:19563809) (part 5-6)

Thank you for bringing this to our attention. We have now included a new sentence with appropriate references in section 6:

"The rupturing capacity of protein VI, as well as its role in regulating capsid stability and endosomal escape have been studied in great detail using a temperature sensitive mutant of human adenovirus 2 and a mutant virus with single point mutation in protein VI, LQ40 (89-93)."

-Maybe it is worthwhile to point out more clearly that endosomal rupture and escape from the endosomal compartment are two separate events (e.g. PMID:22855481) (part 5).

We have included the following sentence in part 7. Escape from endosomes and host detection:

"Specifically, during endosomal breakage and virus escape, which are spatially and temporally distinct events (95), compartmentalized glycans are released to the cytosol where in the foreign context they elicit an autophagic response that is selective in eliminating the damaged components that could become toxic to the cell (see Figure 2 autophagy) (98)."

Responses to Reviewer 2

1. In the Abstract and Conclusions, protein VI appears to be featured among other viral proteins. However, the main text describes general topics on adenoviral transport without specific focus on the protein. Although it is clear that protein VI is a key player in several processes mentioned in the manuscript, the Abstract would need to be more balanced according to the contents of the manuscript.

The abstract has been modified slightly:

There are many remarkable aspects about the AdV entry program, for example, the virus gains targeted control of a large well-defined local network neighborhood by coupling several interacting processes (including endocytosis, autophagy, and microtubule trafficking) around a collective reference state centered on the interactional topology and multifunctional nature of protein VI.

2. I feel some parts of the text are a bit difficult to follow. For instance, page 1 gives a general introduction, page 2 explains specific structural features of adenoviruses, page 3 goes back to a general topic on host-virus interactions, and then page 4 is again about adenoviruses. Reorganizing the text would increase readability.

Thank you for raising this issue and we agree that parts of the text were hard to follow. We have reorganized the manuscript into the following sections:

1. Introduction
2. Cell networks are open but protective against viral infection
3. Adenoviruses are built for cellular networking
4. Receptor binding and initiation of the uncoating program
5. Endocytic uptake
6. Rupturing the endocytic network
7. Escape from endosomes and host detection
8. Cytoplasmic transport (note here that the innate immune sensing description was moved to section 2)
9. Genome uncoating and delivery at the nuclear pore complex
10. Conclusions

3. Subheads are mislabeled: 1. Introduction, 2. Adenoviruses are built for cellular networking, and "2". Cell networks are open but protective against viral infection.

We have fixed the numbering issue.